# The First Case Report of a Solitary Pulmonary Metastasis of a Transitional Meningioma and Literature Review

**DOI:** 10.3390/ijms26146868

**Published:** 2025-07-17

**Authors:** Sara Di Lorenzo, Stefano Farese, Ciro Balbo, Federica Melisi, Marianna Scrima, Lucia Stefania Pasquale, Maria Pasqualina Laudato, Teresa Peluso, Domenico Solari, Andrea Ronchi, Marina Accardo, Renato Franco, Raffaele Addeo, Teresa Somma, Mario Pirozzi, Fortunato Ciardiello, Michele Caraglia, Morena Fasano

**Affiliations:** 1Medical Oncology, Department of Precision Medicine, University of Campania “Luigi Vanvitelli”, 80131 Naples, Italy; stefano.farese93@gmail.com (S.F.); ciro.balbo@studenti.unicampania.it (C.B.); mp.laudato@gmail.com (M.P.L.); fortunato.ciardiello@unicampania.it (F.C.); morenafasano@ymail.com (M.F.); 2Laboratory of Precision and Molecular Oncology, Biogem Scarl, Institute of Genetic Research, Contrada Camporeale, Ariano Irpino, 83031 Avellino, Italy; federica.melisi@biogem.it (F.M.); marianna.scrima@biogem.it (M.S.); luciastefania.pasquale@biogem.it (L.S.P.); teresa.peluso@biogem.it (T.P.); michele.caraglia@unicampania.it (M.C.); 3Division of Neurosurgery, Department of Neurosciences, Reproductive and Odontostomatological Sciences, Federico II University, 80131 Naples, Italy; domenico.solari@unina.it (D.S.); teresa.somma@unina.it (T.S.); 4Pathology Unit, Department of Mental and Physical Health and Preventive Medicine, University of Campania “Luigi Vanvitelli”, 80131 Naples, Italy; andrea.ronchi@unicampania.it (A.R.); marina.accardo@unicampania.it (M.A.); renato.franco@unicampania.it (R.F.); 5Oncology Unit, Hospital “San Giovanni di Dio” of Frattamaggiore, ASL Napoli 2 Nord, 80027 Frattamaggiore, Italy; raffaele.addeo@aslnapoli2nord.it; 6SCDU Oncologia, Azienda Ospedaliera Universitaria Maggiore della Carità, 28100 Novara, Italy; m.mariopirozzi@gmail.com; 7Department of Translational Medicine (DIMET), University of Eastern Piedmont (UPO), 28100 Novara, Italy; 8Department of Precision Medicine, University of Campania “Luigi Vanvitelli”, 80138 Naples, Italy

**Keywords:** lung metastasis, transitional meningioma, beta2-microglobulin, Ataxia–Telangiectasia Mutated

## Abstract

Extracranial metastases from meningiomas are extremely rare, with an incidence of <1%, and their prognosis is poor. Moreover, there is currently no gold standard for their treatment; therefore, the decision-making process is strictly dependent on multidisciplinary discussions. In this report, we describe the case of a 73-year-old patient who was diagnosed with a solitary lung metastasis more than 20 years after the initial treatment for a low-grade meningioma. Molecular characterization of this metastasis was performed using the Oncomine Comprehensive Assay Plus, which identified multiple functional mutations in the beta2-microglobulin (β2M) and ATM genes, both of which may contribute to immune evasion and genomic instability. A short overview of the literature is also reported. To our knowledge, no previous reports exist on single pulmonary metastasis from low-grade meningioma occurring more than 20 years after diagnosis.

## 1. Introduction

Meningioma is the most common primary non-glial brain tumor worldwide [1], constituting around 30% of primary brain tumors and roughly 54% of primary benign ones [2]. Three grades are described in the World Health Organization (WHO) classification [[3],[4],[5][6]]. This tumor is often associated with alterations in the neurofibromatosis type 2 (NF2) gene [5]. Grade 1 tumors are the most common and least aggressive [3]. The majority of meningiomas have a favorable prognosis [7], even if they are locally aggressive. A very small percentage (less than 1%) of any grade meningiomas can metastasize to extracranial sites [5,7,8,9]. Due to the extreme rarity of meningioma metastases, there is a dearth of established management protocols [10].

In this report, we present an interesting case of a patient diagnosed with a single metastatic lung lesion from transitional meningioma with features of microcystic meningioma twenty years after the onset of the primary tumor, with histological features suggestive of an aggressive biological behavior and genetic alterations potentially involved in the modulation of the immune response and genomic stability. This highlights the need for a diagnostic approach that integrates a wide range of molecular profiles for future therapeutic personalization.

## 2. Case Report

In June 2023, a 73-year-old man came to our attention during an outpatient visit.


Clinical history


In November 2002, after an episode of buccal deviation and intense headaches, he underwent brain magnetic resonance imaging (MRI) with contrast, showing an extracerebral neoformation in the right frontal convexity, compatible with meningioma. In March 2003, he underwent surgery. Histological examination reported transitional meningioma with aspects of microcystic meningioma, where the presence of focal areas of necrosis and cytological atypia could be an expression of aggressive biological behavior. There was no residual disease on post-surgery MRI. Thereafter, he underwent clinical and radiological follow-up. In January 2004, a brain MRI showed residue of the previously removed lesion. Therefore, he underwent surgery and, subsequently, adjuvant radiotherapy (RT). Brain MRI after neurosurgery showed a small meningiomatous residue in the right parasagittal frontal region. Then, he was subjected to instrumental follow-up, first every six months and then annually, which confirmed the stability of the meningeal residue. In January 2023, due to the onset of persistent and irritable cough episodes unresponsive to medical therapy, a Computed Tomography (CT) scan was performed, which revealed a 20 × 5 mm nodule in the right upper lobe.

Positron emission tomography with 18-F-fluorodeoxyglucose (18-FDG-PET) showed hyperaccumulation of the tracer in the right lung (SUV max 6.2) and the right frontal lobe (SUV max 15.0). In May 2023, he performed a TRU-CUT needle biopsy of the right lung lesion. The histological examination revealed fragments of lung parenchyma that were the site of proliferation of epitheliomorphic elements with bland morphology, characterized by large cytoplasm with clear margins, organized in solid nests and occasionally in vorticoid structures. Immunohistochemical analysis showed positivity for CK8/18 and CD56, and focally for S100, as observed in the case of meningothelial neoplasia. Chromogranin, synaptophysin, TTF1, P40, PGR, EMA, OLIG2, STAT6, GFAP, CEA, CD15, and ESA were all negative; the expression of BAP1 was retained. The Ki67/MIB1 growth fraction was 3%. The morphological and immunohistochemical picture supported the main possibility as the secondary localization of the known encephalic meningiomatous lesion.


Clinical presentation, Diagnosis, and Treatment


At the outpatient visit, the patient presented with partial orientation in time and space, sporadic episodes of a non-productive cough, and no neurological symptoms. Therefore, we conducted a histological review of the lung aspiration slides and a re-evaluation using instrumental methods. The CT scan confirmed the previously known right lung lesion, which had now slightly increased in size. Meanwhile, the PET scan revealed an increase in hyperaccumulation in the right lower lobe nodule (SUV max 9.2 vs. 6.2) and the right frontal lobe (SUV max 19.3 vs. 15.0). The histological review described a predominantly monomorphic, polygonal epithelioid-type cell population, organized in lobules or the typical “vortex” appearance. The nuclei, rounded and nucleolated, showed focal and slight atypia; the cytoplasm, abundant and eosinophilic, sometimes appeared vacuolized. Mitoses were rare (2 × 10 HPF 40X). The morphological and immunohistochemical picture was compatible with a secondary lesion from meningioma with meningothelial-type aspects. The case was discussed collegially in the multidisciplinary lung cancer oncology group in August 2023, and an atypical lung resection was performed in Video-Assisted Thoracic Surgery (VATS) in October 2023. Histopathological examination revealed medium-sized, monomorphic cell clusters with a syncytial, fasciculated, and whorl-like growth pattern, the presence of psammomatous bodies, and rare mitoses, all characteristics compatible with a secondary lesion from a meningothelial-type meningioma (Figure 1). The patient then underwent follow-up with a total body CT scan, an 18-FDG PET scan, as well as a brain MRI. The last clinical-instrumental revaluation in October 2024 revealed no evidence of disease recurrence. The Figure 2 shows the clinical evolution of the disease during the time (Figure 2).


Materials and Methods and Results


We also performed an Oncomine Comprehensive Assay Plus on the paraffin-embedded lung metastasis tissue to identify potential genetic alterations in the tumor, which may, at least in part, explain the uncommon presentation of the meningioma. The genomic profiling of the meningioma metastasis revealed multiple nucleotide variants (MNVs), dominantly truncated variants, caused by Deletion/Insertions (DELINs) at the microsatellite site c.41-48 of the Beta-2-Microglobulin (B2M) gene. Reported alterations were all nonsense mutations, leading to a truncated protein, and showed different frequency: c.41_42delCTinsAA variant allele frequency (VAF) 80.3%, c.43_45delCTTinsTAG (VAF 19.46%), c.47_48delCTinsAA (VAF 21.48%). Interestingly, we also found a mutation of the Programmed Cell Death 1 Ligand 2 gene (PDCD1LG2) (c.789_791delCAC, VAF 92.97%) of clear functional significance. Moreover, we found a mutation in the Ataxia–Telangiectasia Mutated (ATM) gene affecting its splicing and determining loss of function (c.4777-1G>C, VAF 85.74%). A c.23T>A (VAF 24.53%) mutation of Cadherin 10 (CDH10) was recorded. The AT-Rich Interaction Domain 1B (ARID1B) c.1592_1593insACC mutation was, on the other hand, likely benign and had a low VAF (8.11%), whereas c.2890delG (VAF 23.08%) determined a frame shift mutation likely affecting the function of the coded protein.

A pathologist reviewed all tumor samples isolated from FFPE blocks to estimate the tumor area for dissection and nucleic acid extraction (more than 80%). DNA and RNA were isolated using the MagMAX™ FFPE DNA/RNA Ultra Kit (Applied BiosystemsTM, Thermo Fisher Scientific, Waltham, MA, USA) in a semi-automatic mode on a KingFisher™ Duo Prime magnetic particle processor (Thermo Fisher Scientific, Waltham, MA, USA). DNA and RNA were quantified using a Qubit Fluorometer with Qubit dsDNA HS Assay kit (InvitrogenTM, Thermo Fisher Scientific, Waltham, MA, USA) and Qubit RNA HS Assay kit (InvitrogenTM, Thermo Fisher Scientific, Waltham, MA, USA). A total of 20 ng of DNA and RNA was used as input. The DNA sample was subjected to a deamination reaction using Uracil-DNA Glycosylase–heat labile (Thermo Fisher Scientific, Waltham, MA, USA). OCA+ primer pools were used for library preparation on the Ion Chef (Thermo Fisher Scientific, Waltham, MA, USA) liquid handler, and sequencing was performed using Ion 550™ Chips in the Ion S5™ Sequencer (Thermo Fisher Scientific, Waltham, MA, USA). Data analysis was performed using Ion Reporter™ Software version 5.2.20. The results are summarized in Table 1.

## 3. Discussion

Meningiomas are among the most frequent tumors of the intracranial central nervous system with an incidence of approximately 13–26% [11], far exceeding gliomas [12].

Grade 1 tumors represent 80% of all meningiomas [6] and are most often slow-growing tumors that have a benign course and a lower risk of post-operative recurrence compared to the other two grades [13]. Transitional meningiomas are among the most common subtypes of grade 1 meningioma [14]; they are also known as mixed ones [15] due to their characteristic transitional nature between meningothelial and fibrous types [6].

In the case of grade 1 meningiomas, only complete excision is performed [11]. For subtotal resection or aggressive histology, and grade 2 and 3 tumors, adjuvant radiotherapy usually follows surgical treatment [16].

Extracranial metastases are extremely rare, occurring in only approximately 0.1% of cases [5]. This greatly worsens the prognosis, reducing survival from 88.3% to 66.5% over 5 years [8].

Due to the rarity of this event, a comprehensive understanding of the risk factors for extracranial metastases is not yet established [8]. Often, before the appearance of extracranial metastases, intracranial recurrences occur several times, thus being the only recognized risk factor [7,12]. Local recurrence rates, after total surgery, range from 9 to 32% [16], and, as observed in our case, the reported recurrence rate of transitional meningioma is surprisingly high [6]. The presence of metastases does not constitute a criterion for the WHO classification [17], as they can occur in both grade 1 and higher-grade meningiomas [7]. Several studies with varying results have been conducted to investigate the relationship between extracranial metastasis and histological grade. Kessler et al. reported a higher association with grade 2 meningiomas, while Surov et al. reported a higher association with grade 3 meningiomas [18]. Therefore, the propensity to metastasize cannot be predicted upon grading [7,17].

The average interval between the primary meningioma and first metastases is approximately 6 years [5]. More precisely, Enomoto et al. performed a meta-analysis of 35 articles, highlighting that the average time to the appearance of metastases was 11.0, 5.4, and 2.0 years for grades 1, 2, and 3, respectively [19].

Lung represents the most common site (60%), followed by abdomen and liver (34%), cervical lymph nodes (18%), long bones, pelvis and skull (11%), pleura (9%), vertebrae (7%), central nervous system (7%), and mediastinum (5%) [20].

It seems that the site of origin of primary meningiomas influences the formation of metastases, as parasagittal and falx localizations are most commonly associated with metastasis [5].

Repeated surgical interventions would represent another risk factor [21], due to damage to the blood–brain barrier, which favors the release of tumor cells into the bloodstream [6,22]. Approximately 90% of extracranial metastases described in various studies occur following surgical resection or shunt surgery; in this way, tumor cells flow into the extra-meningeal blood and lymphatic vessels, leading to systemic dissemination [5,12,22].

The metastatic sites depend upon the route of dissemination [12]. It is reported that 75% of patients with extracranial metastases of meningioma present with invasion of the venous sinus [19], which is often accompanied by lung metastases [23]. Another route is that cerebrospinal fluid is naturally exposed to meningiomas for anatomic reasons, allowing them to reach less common metastasis sites [12].

The pathogenesis of meningioma metastases is currently unknown, but it has been observed that a probable predictor of multiple pulmonary metastases is the loss of heterozygosity at 9p, 1p, and 22q [5,12,22]. It has also been reported that CD90 is highly expressed in lung metastasis from meningioma [19].

Meningiomas harboring the NF2 mutation, recorded in numerous cases of metastatic meningioma, appear to have greater genetic instability [6,8].

TRAF7, SMO, AKT1, and KLF4 mutations, referred to as “non-NF2,” are typically found in lower-grade meningiomas, characterized by fewer chromosomal abnormalities, and generally result in better clinical outcomes [24].

In our case, no mutations in these genes were found; however, alterations in the beta-2-microglobulin (B2M) gene sequence were recorded, suggesting a loss-of-function mutation (likely germinal). B2M gene, located on chromosome 15 (15q21.1), encodes a non-glycosylated protein that shares structural similarities with the immunoglobulin (Ig) constant region and the α3 domain of the major histocompatibility complex (MHC) class I molecule [25]. B2M is found both in free form and attached to the cell membrane, and the free form is a significant prognostic factor and predictor of survival in various types of cancer. It is a crucial component of MHC class I molecules, and its alterations can lead to a null or lower expression of the MHC class I complex, which can compromise the mechanism of antigen presentation to the immune cell system [25]. Moreover, B2M exhibits significant alterations in cancer tissues, particularly in tumors characterized by microsatellite instability (MSI) and deficient mismatch repair (dMMR) [26]. Interestingly, the human B2M has dinucleotide repeat regions that are hotspots for frameshift indels, which could alter or silence B2M and thus affect the antigen–presentation process [26]. It is widely recognized that the altered presentation of antigens induced by the alteration or loss of B2M expression is associated with an increased risk of metastasis in various cancers, including melanoma and sarcoma [27,28]. This effect is likely associated with increased immunological escape, as demonstrated recently in a melanoma patient treated with immunological checkpoint inhibitors [29]. Additional studies are warranted to define the real function of this protein.

The programmed Cell Death 1 Ligand 2 (PDCD1LG2) gene encodes the ligand for the surface receptor PD-1 [30], which belongs to the B7 protein family [31] and is expressed on antigen-presenting cells [32]. It plays a primary role in immune tolerance and autoimmunity [33]. Wang reported that PDCD1LG2 in gliomas was closely related to inflammation and immune responses and was associated with better survival [34]. Moreover, mutation of PDCD1LG2 was correlated with lower survival rates in patients with brain metastases from NSCLC who underwent surgery [35]. The function of PDCD1LG2 is not yet fully understood, as its expression has also been recently reported in stromal-derived follicular dendritic cells of lymph node germinal centers, which appear to be involved in the complex processes of T-lymphocyte maturation [36]. However, the PDCD1LG2 mutation identified in our patient has not yet been shown to have clinical significance and warrants further investigation to elucidate its role in the metastatic process. A mutation in the Ataxia–Telangiectasia Mutated (ATM) gene that affects its splicing and causes loss of function [37] has also been found (c.4777-1G>C, VAF 85.74%). This variant has been classified as probably pathogenic, even though its association with cancer was never previously reported [38]. This gene encodes a protein kinase that is recruited in response to DNA damage and phosphorylates numerous proteins involved in DNA repair, cell cycle checkpoints, and apoptosis [33]. Additionally, its mutations are reported to be involved in the tumorigenesis and metastasis of several cancers through SNAIL stabilization and epithelial–mesenchymal transition [39]. Somatic mutations in CDH10, expressed predominantly in the brain and involved in synaptic adhesions and axon growth and guidance [40], are associated with colorectal, gastric, and lung cancer and are considered driver mutations in pancreatic cancer [41]. Moreover, its mutations have been frequently reported in patients with lung squamous cell cancer, suggesting a role in tumor suppression and cell–cell interaction, thereby supporting evidence of its involvement in metastasis formation [42].

The AT-Rich Interaction Domain 1B (ARID1B) c.1592_1593insACC mutation was likely benign and had a low VAF (8.11%), whereas c.2890delG (VAF 23.08%) determined a frame shift mutation likely affecting the function of the coded protein. ARID1B is a DNA-binding subunit of the Brahma-associated factor chromatin remodeling complex, engaged in the regulation of gene activity. Its mutation has been found in neurodevelopmental disorders [43]. This mutation, too, was never previously associated with cancer development.

Then, excluding B2M and ATM, regarding the other mutations found in the other genes, no definitive conclusions can be drawn about their role in the development of meningioma due to the lack of previous evidence.

An important support for neuropathologists in determining aggressiveness is the nuclear antigen expressed during the active phases of the cell cycle, called Ki-67 [17]. Previous research has reported a high proliferative index of Ki-67 (MIB-1) as one of the factors that also increases the risk of metastasis [6]. Typically, the proliferative index tends to increase in correlation with the grade. More precisely, benign meningiomas have an average proliferative index of approximately 4%, while anaplastic ones have an index of approximately 14.7% [6]. In our case, the Ki-67/MIB-1 growth fraction of the metastatic lesion was 3%, which is equal to that of the primary tumor. This finding makes our case even more peculiar.

Currently, there is no standard protocol for managing meningioma metastases [24]. Surgical or stereotactic radiosurgical resection improves survival, although the literature is very limited [24]. Chemotherapy with hydroxyurea, temozolomide, and trabectedin has limited efficacy and is not recommended, as it has achieved poor results in terms of both disease control and survival [44]. Several ongoing clinical trials are investigating systemic therapies in combination with immunotherapy, 177Lu-DOTATATE, tyrosine kinase inhibitors, abemaciclib, capivasertib, and vismodegib [24].

## 4. Conclusions

In conclusion, we presented a case of low-grade meningioma with lung metastasis 20 years after the initial surgery. Although the histological characterization indicated a grade 1 and a low Ki-67%, it presented aggressive pathological features, which may be associated with the development of both local recurrence and distant metastases. Interestingly, new functional mutations were identified in the B2M and ATM genes. The best approach for patients with this diagnosis has yet to be determined.

Further clinical studies are needed to define the pathogenesis and genetic alteration of extracranial metastases. For example, multicenter retrospective cohort studies are essential to collect a sufficient number of metastatic cases to identify undefined clinical and histopathological characteristics. Meanwhile, prospective cohort studies could also provide information on predictors of aggressive behavior. New molecular profiling technologies, such as next-generation sequencing (NGS), combined with liquid biopsy or artificial intelligence (AI)- based radiomics imaging data, could be used to predict the risk of recurrence. Furthermore, single-cell sequencing and study of the tumor microenvironment could further define the cellular subpopulations and interactions that determine their dissemination. Support could come from the creation of biobanks that facilitate the sharing of high-quality, standardized images, as well as comprehensive clinical, pathological, and molecular data. The integration of these data with real-world data could revolutionize both the management and treatment of metastatic meningioma using innovative drugs.

## Figures and Tables

**Figure 1 ijms-26-06868-f001:**
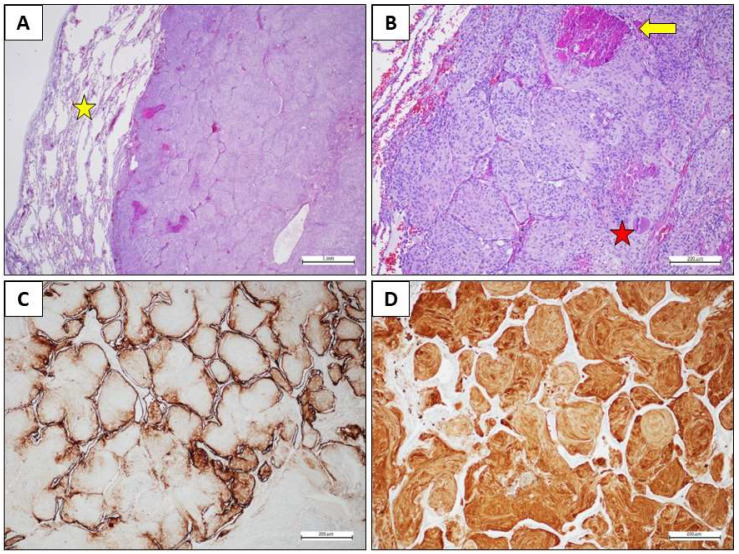
Histological findings. (**A**) Histological examination reveals a solid neoplastic mass within the lung parenchyma (yellow star), exhibiting a pushing, expansive growth pattern (Hematoxylin and eosin stain, original magnification 20×, scale bar = 1 mm). (**B**) The tumor shows a lobular architecture and is composed of elongated cells with eosinophilic cytoplasm, oval nuclei, and inconspicuous nucleoli. An eosinophilic globule (red star) and areas of necrosis (yellow arrow) are observed (Hematoxylin and eosin stain, original magnification 100×, scale bar = 200 µm). (**C**) EMA immunohistochemistry showed faint positivity in a few neoplastic cells, with strong expression in pneumocytes (immunohistochemical stain, original magnification 100×, scale bar = 200 µm). (**D**) S100 immunohistochemistry revealed diffuse positivity in the neoplastic cells (immunohistochemical stain, original magnification 100×, scale bar = 200 µm).

**Figure 2 ijms-26-06868-f002:**
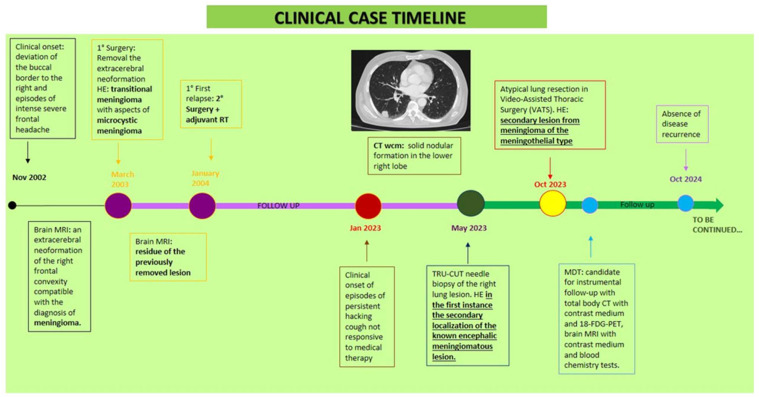
Clinical case timeline. Abbreviations: RT: radiotherapy; CT: Computed Tomography; WCM: with contrast medium; HE: histological examination; MRI: Magnetic Resonance Imaging; MDT: multidisciplinary team.

**Table 1 ijms-26-06868-t001:** Nucleic acid isolation and sequencing.

GENE	CODING	PROTEIN	dbSNP	VAF %
CDH10	c.23T>A	p.Leu8Gln		24.53
ARID1BꟾMIR4466	c.1592_1593insACCꟾ	p.Pro533dupl	rs572236007:rs1405399488	8.11
PDCD1LG2	c.789_791delCAC	p.Thr265del	rs1816600126:rs890047189	92.97
ATM	c.4777-1G>C	p.?	rs1591684268	85.74
B2M	c.41_42delCTinsAA	p.Ser14Ter	rs986783978:rs1341857550	80.30
B2M	c.43_45delCTTinsTAG	p.Leu15Ter	rs1329285364	19.46
B2M	c.47_48delCTinsAA	p.Ser16Ter	rs1435611656	21.48
CREBBP	c.2890delG	p.Ala964GlnfsTer34	rs760906604	23.08

*Abbreviations*: CDH10: Cadherin 10; ARID1B: AT-Rich Interaction Domain 1B; PDCD1LG2: Programmed Cell Death 1 Ligand 2; ATM: Ataxia–Telangiectasia Mutated; B2M: Beta-2-Microglobulin; CREBBP: cAMP response element-binding protein.

## Data Availability

Data are contained within the article.

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
