# Peer review of "The First Case Report of a Solitary Pulmonary Metastasis of a Transitional Meningioma and Literature Review"

_ijms, 2025, doi:10.3390/ijms26146868_

Round 1

Reviewer 1 Report

Comments and Suggestions for Authors

This is a very rare case of a male patient in his 70´s with a metastatic meningioma. Tee case is very sound and pathological/molecular testing was done to confirm diagnosis. I would suggest adding a CRediT statement; as a case report it is hard to understand the role of 18 authors. 

Minor English editing errors: "...he performed brain magnetic resonance" he did no do a MRI to himself...

Comments on the Quality of English Language

Some minor English Editing errors were found

Author Response

Comment 1: I would suggest adding a CRediT statement;

Response: We've added them.

  Comment 2: Minor English editing errors Response: We have taken steps to improve it  

Reviewer 2 Report

Comments and Suggestions for Authors

The authors present an interesting case of pulmonary metastasis from a low grade meningioma.

-The authors present tumor genetics for the lung metastasis which reveals several interesting variants. Was this also done on archived tissue from the primary resection? It would be interesting to see how the two sites compare. 
-Did the PET scan show any other sites of disease aside from the lung nodule? It would be nice to see a whole body PET image in the figures. 
-Did the patient have any lung imaging between initial meningioma diagnosis and current presentation? It would be interesting to know how long the lung nodule was detectable on imaging. 

Author Response

-Comment 1:The authors present tumor genetics for the lung metastasis which reveals several interesting variants. Was this also done on archived tissue from the primary resection? It would be interesting to see how the two sites compare. 

Response1: We agree with the referee regarding the interest in a possible concordance between the gene mutations found in the metastasis and the primary tumor. However, the primary bioptic specimen was used entirely for the initial diagnosis, and no additional slides were available for further molecular studies.

-Comment 2:Did the PET scan show any other sites of disease aside from the lung nodule? It would be nice to see a whole-body PET image in the figures. 

Response2 :Yes, the PET scan revealed an additional site of pathological hyperuptake in the brain, in addition to the pulmonary nodule. Specifically, an area of ​​hypermetabolism was confirmed in the right frontal region, in the mid-superior region (maximum SUV 19.3), with a heterogeneous morphology. This area appears to partially circumscribe a hypometabolic malarial area of ​​ipsilateral frontobasal hypometabolism in the frontobasal region, consistent with the results of previous neurosurgery. Purtroppo non disponiamo immagini della PET a corpo intero.

-Comment 3:Did the patient have any lung imaging between initial meningioma diagnosis and current presentation? It would be interesting to know how long the lung nodule was detectable on imaging. 

Response 3: Unfortunately, the clinical records do not show any chest imaging tests performed before January 2023 (which coincides with the onset of the cough).

Reviewer 3 Report

Comments and Suggestions for Authors

This manuscript presents a well-documented and clinically significant case report describing what appears to be the first documented instance of a solitary pulmonary metastasis from a WHO grade 1 transitional meningioma, occurring more than 20 years after the initial diagnosis and treatment. The topic is both relevant and novel, offering valuable insights into the rare phenomenon of late extracranial dissemination in low-grade meningiomas.

Suggestions: 

1- The report addresses a highly unusual presentation, adding valuable data to the limited literature on metastatic behavior in grade 1 meningiomas. This should be highlighted more explicitly in the abstract and introduction to better frame the novelty of the case.

2- The authors should explore potential mechanistic links between these alterations and the metastatic behavior observed.

3- Mutations in genes such as PDCD1LG2 and CDH10 are mentioned but not adequately contextualized. Clarify whether these are of known or unknown clinical significance and consider referencing relevant studies if available.

4- improve English. 

5- Figure 1 would benefit from clearer labeling and the inclusion of scale bars. 

6-Reference formatting should be standardized throughout the manuscript.

7- “Frustules of lung parenchyma” is likely an incorrect term. Consider using “fragments” or “specimens.”

8- In the conclusion, avoid vague statements such as “we need more studies.” Instead, specify what types of studies would be most valuable (e.g., prospective cohorts, molecular profiling initiatives).

Comments on the Quality of English Language

English needs improvement throughout the text.

Author Response

Comment 1:The report addresses a highly unusual presentation, adding valuable data to the limited literature on metastatic behavior in grade 1 meningiomas. This should be highlighted more explicitly in the abstract and introduction to better frame the novelty of the case.

Response: We have taken steps to modify it.

2- Comment 2:The authors should explore potential mechanistic links between these alterations and the metastatic behavior observed.

3-Comment 3:Mutations in genes such as PDCD1LG2 and CDH10 are mentioned but not adequately contextualized. Clarify whether these are of known or unknown clinical significance and consider referencing relevant studies if available.

Response 2-3: We have expanded the discussion about the gene mutations found in the lung metastasis and added in the "Discussion" section of the revised version of the manuscript the following statements: "It is widely known that the affected presentation of antigens induced by alteration/loss of expression of B2M is associated with an increased risk of metastasis occurrence in different cancers including melanoma and sarcoma [PMID: 30684558; PMID: 39249578]. This effect is likely associated with increased immunological escape as demonstrated recently in a melanoma patient treated with immunological checkpoint inhibitors [PMID: 29070816]. The programmed Cell Death 1 Ligand 2 (PDCD1LG2) gene encodes the ligand for the surface receptor PD-1 [29], which belongs to the B7 protein family [30] and is expressed on antigen-presenting cells [31]. It plays a primary role in immune tolerance and autoimmunity [32]. Wang reported that PDCD1LG2 in gliomas was closely related to inflammation and immune response, and was associated with better survival [33]. Moreover, mutation of PDCD1LG2 was correlated with lower survival rates in patients with brain metastases from NSCLC who underwent surgery [PMID: 30218756]. The function of PDCD1LG2 is not yet fully understood, as its expression has also been recently reported in stromal-derived follicular dendritic cells of lymph node germinal centers, suggesting involvement in the complex processes of T-lymphocyte maturation [PMID: 34424268]. However, the PDCD1LG2 mutation found in our patient does not yet have clinical significance and warrants additional investigations to determine its role in the metastatic process. A mutation in the Ataxia-Telangiectasia Mutated (ATM) gene that affects its splicing and causes loss of function [34] has also been found (c.4777-1G>C, VAF 85.74%). This variant has been classified as probably pathogenic, even though its association with cancer was never previously reported [35]. This gene encodes a protein kinase that is recruited in response to DNA damage and phosphorylates numerous proteins involved in DNA repair, cell cycle checkpoints, and apoptosis [36]. Additionally, its mutations are reported to be involved in the tumorigenesis and metastasis of several cancers through SNAIL stabilization and the epithelial-mesenchymal transition [PMID: 38981349]. A missense mutation of the Cadherin 10 (CDH10) gene was also recorded with no previous association with cancer. Somatic mutations in CDH10, expressed predominantly in the brain and involved in synaptic adhesions and axon growth and guidance [37], are associated with colorectal, gastric, and lung cancer and are considered driver mutations in pancreatic cancer [38]. Moreover, its mutations were frequently reported in lung squamous cell cancer patients, suggesting its role in tumour suppression and cell-cell interaction, thus supporting the evidence on its involvement in metastasis formation [PMID: 26503331]." In the previous statements, the role of PDCD1LG2 and CDH10 gene mutations is also adequately contextualized. The clinical significance of each mutation was also clearly stated.

4- Comment 4:improve English. Response: We have taken steps to modify it.

5-Comment5: Figure 1 would benefit from clearer labeling and the inclusion of scale bars. 

Response: As appropriately suggested by the referee, we have added more explicit labeling and scale bars to Figure 1 of the revised manuscript.

6-Comment 6:Reference formatting should be standardized throughout the manuscript.

Response6: We have taken steps to modify it.

7-Comment: "Frustules of lung parenchyma" is likely an incorrect term. Consider using "fragments" or "specimens."

Response 7: As correctly suggested by the referee, we have replaced "Frustules" with "Fragments".

8-Comment8: In the conclusion, avoid vague statements such as "we need more studies." Instead, specify what types of studies would be most valuable (e.g., prospective cohorts, molecular profiling initiatives).

 Response8: Further clinical studies are needed to define the pathogenesis and genetic alteration of extracranial metastases. For example, multicenter retrospective cohort studies are essential to collect a sufficient number of metastatic cases to identify unknown clinical and histopathological features. Meanwhile, prospective cohort studies could also provide information on predictors of aggressive behavior. New molecular profiling technologies such as next-generation sequencing (NGS), followed by the use of liquid biopsy or radiomics, could be used to predict the risk of recurrence. Furthermore, single-cell sequencing and the study of the tumor microenvironment could further define the cellular subpopulations and interactions that determine their dissemination. Support could come from the creation of biobanks that facilitate the sharing of high-quality, standardized imaging, as well as comprehensive clinical, pathological, and molecular data. The integration of this data with real-world data could revolutionize both the management and treatment of metastatic meningioma with innovative drugs.

Reviewer 4 Report

Comments and Suggestions for Authors

The authors describe a clinical case of metastatic meningioma. Extracranial metastases of meningioma are rare and their prognosis is unfavorable. No full knowledge of risk factors, no established management protocols and gold standards for the treatment of these cases and each case is important to study.

All necessary sections for case rerort in total are present, the authors describe the background and context of this case report, the patient information and history, current examination findings, intervention detailed methods and diagnostic evaluations, the explanation of the importance and relevance; the informed consent from the patient is present.

 New data on the detected mutations and the contradictory behavior of the tumor in the context of the proliferative index of Ki-67 are interesting and important for future research, but not yet for use in clinical practice.

The paper is well written, the text clear to read. The statements and conclusions drawn coherent and supported by the citations. This article can be accepted for publication after minor revision.

  1. Please clarify whether there were any studies in 2003-2004 other than MRI and histological examination? Immunohistochemical analysis and Ki67 - only in 2023? If any molecular studies were performed in both 2003-2004 and 2023, please describe whether there were any differences.

Author Response

  1. Comment 1:Please clarify whether there were any studies in 2003-2004 other than MRI and histological examination? Immunohistochemical analysis and Ki67 - only in 2023?

Response 1: We do not have sufficient evidence to confirm the performance of further diagnostic tests in the period 2003–2004, beyond MRI and histological examination. Specifically, no data are available relating to immunohistochemical studies or determination of the proliferative index.

  1. Comment 2: If any molecular studies were performed in both 2003-2004 and 2023, please describe whether there were any differences.

Response 2: As in the case of the concern raised by Referee #1, we agree with the referee regarding the interest in a possible concordance between the gene mutations found in the metastasis and primary tumor. However, the primary histological specimen was used entirely for the initial diagnosis, and no additional slides were available for further molecular studies. However, we have added the Ki-67 score of the primary tumor (as indicated in our records) to the text of the "Discussion" section in the revised version of the manuscript. As you can see, the score of the primary tumour was the same as that of the metastatic tissue.